# Knowledge and determinants of infection prevention and control compliance among nurses in Yendi municipality, Ghana

Abdul-Manaf Mutaru[1,2]*, Augustine Ngmenemandel Balegha[2,3], Raymond Kunsu[1], Collins Gbeti[4]

1 Department of General Nursing, College of Health Sciences, Yendi, Northern Region, Ghana,
2 Department of Behavioural and Social Change, School of Public Health, University for Development Studies, Tamale, Northern Region, Ghana, 3 Department of Obstetrics and Gynaecology, Upper West Regional Hospital, Wa, Upper West Region, Ghana, 4 Department of Health Science Education, University for Development Studies, Tamale, Ghana

* abdulmanafmutaru@gmail.com

## Abstract

### Background

Knowledge of and compliance to infection prevention and control (IPC) guidelines are crucial to curbing hospital acquired infections (HAIs). Globally, about 7–10% of patients suffer HAIs. However, there is limited evidence on nurses' knowledge and adherence to IPC guidelines. Therefore, this study assessed the knowledge and determinants of IPC compliance among nurses in Yendi Municipality, Northern Ghana.

### Methods

We conducted a quantitative cross-sectional study among 209 nurses of the Yendi Municipal hospital, using an adapted questionnaire. We collected and analysed data using SPSS version 26. Socio-demographics, knowledge level and compliance to IPC guidelines were assessed using descriptive statistics. The minimum scores for knowledge and compliance were 0 each with maximum scores being 10 and 8 respectively. Using binary multivariate logistic regression, the determinants of IPC compliance were analysed and odds ratios reported at 95% confidence intervals.

### Results

The nurses had high mean and standard deviation scores for knowledge (7.26 ± 1.4) and compliance to IPC guidelines (5.41 ± 1.5). Females (aOR: 0.33; 95%CI: 0.17–0.64; p = 0.001) were significantly less likely to comply to IPC guidelines. Nurses working in Maternity/Labour wards (aOR: 8.31; 95%CI: 2.46–28.15; p = 0.001) and Outpatient Department (OPD)/Psychiatry (aOR: 5.00; 95%CI: 1.42–17.62; p = 0.012) were associated with higher odds of complying to IPC guidelines. Availability of IPC guidelines (aOR: 3.48; 95%CI: 1.13–10.72; p = 0.030) in a working department influenced compliance to IPC measures.

**Data Availability Statement:** All relevant data are within the paper and its Supporting information files.

**Funding:** The authors received no specific funding for this work.

**Competing interests:** The authors have declared that no competing interest exist.

## Conclusion

The study revealed high knowledge and high compliance regarding IPC among nurses in the Yendi Municipal Hospital. A person's working department and the availability of IPC guidelines were key determinants for high compliance to IPC measures. However, knowledge of IPC did not influence compliance to IPC guidelines. The Municipal and Regional Health directorates, should therefore ensure adequate logistic flow, provision of IPC guidelines and proper supervision to ensure maximum compliance with IPC guidelines, particularly among paediatric, medical and surgical wards nurses as well as degree holding nurses.

## Introduction

Hospital Acquired Infections (HAIs) are serious public health problems in both industrialized and developing economies, causing great morbidity, mortality and associated with high expenditure [1]. HAIs are occupation related infections among hospital staff or infections acquired by patients by virtue of their being hospitalized [1]. Globally, HAIs account for 7–10% of infections among patients. An estimated 30% of patients in the intensive care units of high-income nations are infected with at least one HAI [1,2]. Infections acquired in intensive care units are at least 23 times more common in low- and middle-income countries than in high-income nations [1]. The WHO Patient Safety initiative is aimed at ensuring efficient ways of improving global health care and protecting lives lost to HAIs. In this initiative, the *Clean Care is Safer Care* programme targets decreasing HAIs globally, with particular attention to hand hygiene practices [3,4]. Furthermore, the United Nations' sustainable development goal three is aimed at ensuring and promoting healthy lives for all at all ages. To achieve this goal, it has been projected that, epidemics of infections including HAIs must be eradicated by 2030 [5,6].

Except in epidemics, preventing HAIs rarely attract significant attention [7]. Available literature suggests that the burden of HAIs is high and widespread in Sub-Saharan Africa, with low vaccination rates against vaccine preventable HAIs [8]. In south east Ethiopia, one out of every 31 hospital patients has at least one HAI on any given day [9]. Most of these illnesses can easily be prevented with appropriate infection prevention and control (IPC) strategies. IPC refers to the practical, evidence-based approach which prevents patients and health workers from being harmed by avoidable infection, including infections arising from antimicrobial resistance [10,11]. However, across many middle income nations, inadequate and inefficient IPC programs account for failure in the successful fight against this menace [12]. Despite the awareness that prevention and control of infection are important, low compliance with standard precautions continues to be a problem in many hospitals [13]. Sahiledengle et al. [9] suggest that adoption of constant supervision is very critical to increasing adherence to infection control methods among workers.

The incidence of HAIs in Ghana continues to rise, despite several interventions [14]. There is an increasing need for health workers to take responsibilities to break the chain of infection in healthcare settings. Through adherence to IPC guidelines, health workers can play critical roles in preventing HAIs [7]. In relatively rural parts of the world like Yendi, northern Ghana, there is limited data regarding clinical nurses' knowledge and compliance to IPC guidelines. Also, inadequacy of equipment, lack of incentives and poor human resource has been highlighted by Bassoumah et al. [15] in their recent study to have characterised the Yendi

municipality's health system. Therefore, this study aims to determine the level of knowledge and related factors on IPC compliance among nurses in the Yendi Municipality, northern Ghana. The results of this study may inform policy makers in the improvement of universal and institutionalized IPC protocols as well as strategies to enhance the knowledge of healthcare workers on IPC as well as HAIs.

## Materials and methods

### Study design and site

This was a cross-sectional survey with quantitative approach, and was conducted in December 2021. The setting was Yendi Municipality, located off the Eastern corridor road of Northern Ghana. According to the 2020 Population and Housing Census, Yendi Municipality has a total population of 154,421, out of which 76,142 are males and 78,279 are females. The Municipality has four health centres, two clinics, four Community-based Health and Planning Services (CHPS) compounds and a Municipal Hospital. The Municipal Hospital serves as a referral centre for the whole municipality. This study was carried out among nurses of the hospital. The total nurse population of the municipal hospital is estimated to be 360 people. The study population included all the various cadre of clinical nurses in the hospital, who were fully employed. The cadre of nurses varies according to the professional training acquired (academic and clinical). Thus, enrolled nurses hold certificate in nursing assistance after receiving a two-year training. Likewise, staff nurses hold a diploma in nursing after receiving a three-year training. Nursing/Midwifery officers hold a bachelor's degree in nursing after receiving a four-year training from the university.

### Sample size determination

The sample size was estimated using Yamane's [16] formula of sample size estimation;

$$n = \frac{N}{1 + Ne^2}$$

Where, n: sample size, N: estimated study population, and e: margin of error. The hospital has an estimated nurse population of 360. At a confidence level of 95% and margin of error of 5%, the sample size was estimated to be 190 participants. The researchers anticipated 10% non-response rate. The final sample size was therefore estimated as 209.

**Inclusion and exclusion criteria.** The inclusion criteria for this study considered professional nurses of all cadre, who were licenced and permanently employed in the facility. However, student nurses on clinical attachment and all nurses on national service were exempted from participating in the study.

**Sampling procedure.** Multistage stratified sampling technique was employed in the selection of participants. First, the nurses were stratified based on department into In-patient and Outpatient departments (OPD). The Outpatient department comprised the Psychiatry, Eye, vital signs/Consultation, and public health units while the Inpatients department consisted of the Medial, Surgical, Maternity/Labour, Surgical operating room (Theatre), Accident and Emergency (A & E), Intensive Care Unit and Paediatrics units. Nurses in each unit of department were further stratified by professional cadre (less than or Principal Enrolled Nurse equivalent [≤ PEN], Senior or Staff Nurse/Midwife [SSNM] and Nursing/Midwifery Officer [NMO]). The details of the sampling strategy, including the probability proportional to strata size approach that we applied, have been presented in Table 1.

**Table 1. Selection based on probability proportional to strata size approach.**

| Professional cadre | Population per cadre | Percentage (%) | Proportional sample size |
| --- | --- | --- | --- |
| Principal Enrolled Nurse (OPD) | 20 | 5.6 | 12 |
| Senior or Staff Nurse/Midwife (OPD) | 28 | 7.8 | 16 |
| Nursing/Midwifery Officer (OPD) | 5 | 1.4 | 3 |
| Principal Enrolled Nurse (IPD) | 96 | 26.7 | 56 |
| Senior or Staff Nurse/Midwife (IPD) | 183 | 50.8 | 106 |
| Nursing/Midwifery Officer (IPD) | 28 | 7.8 | 16 |
| **Total** | **360** | **100** | **209** |

OPD-Outpatient department; IPD- Inpatient department.

## Data collection

The data collection tool was a pretested self-administered structured close ended questionnaire. The questionnaire (S1 Questionnaire) was adapted from previously published studies [9,14,17] and revised to suit the objective of the study. The Questionnaire included 28 questions and structured into three main sections. Section A; Socio-demographic characteristics of respondents, elicited responses on respondents' age, gender, professional cadre, working department, qualification, years of practice, having prior IPC training, availability of IPC committee, use of personal protective equipment (PPEs) and IPC guidelines in working department. Section B used ten questions regarding assessment of knowledge on IPC to elicit "Yes or No" responses from participants. The knowledge level of respondents was assessed on components of IPC- disease prevention, training, management, logistics. Section C which comprised eight questions assessed compliance to IPC guidelines, using "Yes or No" to elicit responses. Compliance with IPC was measured in respect of hand washing, disinfection, cross infection and availability of IPC guidelines.

After obtaining written informed consent, hard copies of questionnaire were given to consented participants through their respective department heads. The study participants were allowed at most 24 hours to respond and anonymously submit completed questionnaire back to the researchers through their head of departments. Completed questionnaire were assessed daily to ensure validity and accuracy of responses.

## Data analysis

Data was sorted, coded and entered into SPSS version 26 for analysis. S2 Data, contains the minimum data set from the survey. Frequencies and percentages were used to summarize data on the socio-demographic characteristics, knowledge and compliance to IPC guidelines. Composite scores for knowledge and compliance to IPC guidelines were computed by scoring the responses of the respondents by summation. Correctly answered questions were each awarded 1 point while wrong answers attracted no point. The minimum score for each of knowledge and compliance to IPC was 0 while the maximum composite knowledge score obtainable by a respondent was 10 and that for compliance to IPC guidelines was 8. Summary statistics of mean (and standard deviation), minimum and maximum of the composite scores on knowledge of IPC and compliance to IPC guidelines were then computed. However, the data for the composite scores on knowledge of IPC and compliance to IPC guidelines were each of normal distribution. Therefore mean (and standard deviation) was reported as the measure of central tendency for each of knowledge and compliance to IPC guidelines. The overall knowledge and compliance to IPC guidelines by the respondents were then categorized based on composite

scores. Using the 50% midpoint of the obtainable composite scores, overall knowledge was categorized into low knowledge of IPC (composite scores of $\leq 5$) and high knowledge (composite scores $> 5$). Likewise, compliance to IPC guidelines was categorized into low compliance (composite scores $\leq 4$) and high compliance (composite scores $> 4$).

Logistic regression models were used to analyse the association between the independent variables (age, gender, rank, department, qualification, years of experience, prior IPC training, availability of IPC committee, availability of PPEs and knowledge of IPC) and dependent variable (compliance to IPC guidelines). In line with published literature, including Chowdhury and Turin [18] and Abubakari et al. [19], bivariate analysis was performed to identify the independent predictors of compliance to IPC guidelines at 20% significance level (p < 0.20). Variables found to be statistically significantly associated with compliance to IPC guidelines were then included in to a multivariate binary logistic regression model to eliminate spurious predictors at 5% significance level, with adjusted odds ratios (aOR) reported at 95% confidence intervals (CI).

## Validity and reliability

The data collection tool, after extensive review of literature was adapted from published works [9,14,17]. Face and content validity of the tool was done and unclear statements were rephrased, after pretesting (S1 Data). We cross-checked the data collected to eliminate ambiguous responses which may allude to guessing. This serves to eliminate response and recall bias which ultimately improves the internal validity of the study. This was performed among participants of similar socio-demographic characteristics and working in other health facilities within the Municipality. Using Cronbach's alpha test for internal consistency reliability, the scales of knowledge on IPC and compliance to IPC guidelines were tested for internal consistency reliability. The alpha coefficients for knowledge and compliance scales were $\boldsymbol{\alpha_k} = 0.691$ and $\boldsymbol{\alpha_c} = 0.783$ respectively, with an overall alpha coefficient of $\boldsymbol{\alpha_o} = 0.761$. These were considered acceptable [20,21].

## Ethical consideration

This study was granted approval by the Committee on Human Research Publications and Ethics of the Kwame Nkrumah University of Science and Technology/School of Medical Sciences (CHRPE/AP/606/21). Institutional access was granted by the Northern Regional Health directorate and the medical superintendent of the Yendi Municipal Hospital. Participants needed to grant consent including the use of data, before being allowed to participate. The study participants were assured of utmost confidentiality regarding the use and storage of the data collected.

## Results

### Socio-demographic characteristics of respondents

From Table 2, the study included 209 participants with a 100 percent response rate, and majority (50.2%) of respondents either worked in Paediatric, Medical or Surgical department of the hospital. Nearly half (48.3%) of the respondents possessed the diploma nursing qualification and most (46.4%) of them being between the ages of 26 and 30 years, with a mean age and standard deviation of 28 ± 4.6 years. Females constituted the majority (50.2%) of the respondents. In terms of professional cadre, nurses with Staff/Senior Staff Nurse/Midwife ranks (58.4%) constituted the majority of respondents. The. Majority (79.4%) of the nurses had worked for about 1–5 years. Most (86.6%) of respondents have had prior IPC training. About

**Table 2. Socio-demographic characteristics of respondents (n = 209).**

| Variables | Frequency | % |
|---|---|---|
| **Age (years)** | | |
| 20–25 | 68 | 32.5 |
| 26–30 | 97 | 46.4 |
| ≥ 31 | 44 | 21.1 |
| Mean (SD) | 28 ± 4.6 | |
| **Gender** | | |
| Male | 104 | 49.8 |
| Female | 105 | 50.2 |
| **Professional Rank** | | |
| ≤ PEN | 68 | 32.5 |
| SSNM | 122 | 58.4 |
| NMO | 19 | 9.1 |
| **Department** | | |
| Theatre/A&E/ICU | 33 | 15.8 |
| Paediatric/Medical/Surgical | 105 | 50.2 |
| Maternity/Labour | 40 | 19.1 |
| OPD/Psychiatry | 31 | 14.8 |
| **Qualification** | | |
| Certificate | 87 | 41.6 |
| Diploma | 101 | 48.3 |
| Degree | 21 | 10.0 |
| **Years of practice** | | |
| 1–5 years | 166 | 79.4 |
| 6–10 years | 43 | 20.6 |
| **Having prior IPC training** | | |
| No | 28 | 13.4 |
| Yes | 181 | 86.6 |
| **Availability of IPC committee** | | |
| No | 38 | 18.2 |
| Yes | 171 | 81.8 |
| **Availability of PPEs** | | |
| No | 42 | 20.1 |
| Yes | 167 | 79.9 |
| **Availability of IPC guideline** | | |
| No | 16 | 7.7 |
| Yes | 193 | 92.3 |

SSNM; Staff/Senior Staff Nurse/Midwife, PEN; Principal Enrolled Nurse, NMO; Nursing/Midwifery Officer, A&E; Accident and Emergency, ICU; Intensive Care Unit, OPD; Outpatient Department.

81.8% of the nurses reported to have an IPC committee available in the facility. The majority (79.9%) of them, reported the availability of PPEs in the facility. Also, 92.3% confirmed the availability of IPC guide in their working departments.

## Knowledge and compliance to IPC guidelines

**Knowledge of respondents on IPC.** Table 3 presents the knowledge of the respondents on IPC. All the respondents (100.0%) were able to confirm the need for hand washing before

**Table 3. Knowledge of respondents on IPC guideline (n = 209).**

| Variables | Frequency (%) | |
|---|---|---|
| | **Yes** | **No** |
| Routine hand hygiene | 209 (100.0) | 0.0 (0) |
| Gloves provides complete protection | 82 (39.2) | 127 (60.8) |
| Recapping needles after injection | 150 (71.8) | 59 (28.2) |
| Antiseptic effectiveness compared to hand washing | 160 (76.6) | 49 (23.4) |
| Use of gloves in blood/fluid exposure | 173 (82.8) | 36 (17.2) |
| Segregation of waste at generation point | 114 (54.5) | 95 (45.5) |
| Tuberculosis (TB) transmission route | 193 (92.3) | 16 (7.7) |
| Change of gloves between patients | 71 (34.0) | 138 (66.0) |
| Preparation of 0.5% chlorine solution | 177 (84.7) | 32 (15.3) |
| Use of Safety box | 178 (85.2) | 31 (14.8) |

and after every procedure. The majority (60.8%) of respondents could not confirm that gloves do not provide complete protection. Less than one third (28.2%) of the respondents reported aptly that all needles should not be recapped after injection. The majority (76.6%) of the respondents confirmed that alcohol hand rub can be effective when hands are not visibly soiled. Most (82.8%) of the respondents reported the need to wear gloves in anticipation of blood/fluid exposure. Also, less than half (45.5%) of the respondents reported that waste should be segregated at the point of generation. Most (92.3%) of the respondents confirmed that TB is carried in air, from an active TB patient. In this study, 66.0% of respondents reported the need to change gloves between patients. Also, 84.7% of respondents knew how to prepare 0.5% chlorine solution. The majority (85.2%) of them reported that safety box should not be used when three-quarters full.

**Compliance with IPC among respondents.** As indicted in Table 4, the majority (87.1%) of respondents reported that they always wash hands before and after patient care. About 68.9% confirmed washing hands with soap under running water for 40 to 60 seconds. Also, 67.9% reported wearing a face mask when attending to clients. Three quarters (75.6%) of the respondents indicated that they considered every patient potentially infectious. More than half (55.5%) of the respondents reported not to use alcohol hand rub after removal of gloves. Only 35.4% of respondents reported not recapping needles before disposing them off. About 69.4% of the respondents reported their disposal of contaminated materials into impermeable bag. Majority (86.1%) of respondents reported using IPC guide in their working departments.

**Table 4. IPC compliance among respondents (n = 209).**

| Variables | Frequency (%) | |
|---|---|---|
| | **Yes** | **No** |
| Do you wash hands before and after patient care | 182 (87.1) | 27 (12.9) |
| Hand washing for at least 1 minute | 144 (68.9) | 65 (31.1) |
| Disposable face mask use when attending to clients | 142 (67.9) | 67 (32.1) |
| Every client is potentially infectious | 158 (75.6) | 51 (24.4) |
| Use of alcohol hand rub after removal of gloves | 93 (44.5) | 116 (55.5) |
| Recapping needles before disposing them off | 135 (64.6) | 74 (35.4) |
| Disposing potentially contaminated materials | 145 (69.4) | 64 (30.6) |
| Evidence of infection prevention and control in practice | 180 (86.1) | 29 (13.9) |

**Overall IPC knowledge level and compliance of respondents.** The overall knowledge level of the nurses on IPC was high. The majority (90.9%) of respondents had high knowledge on IPC, with a mean composite score of 7.26 ± 1.4. The minimum and maximum composite scores were 2 and 10 respectively. With compliance, more than half (65.6%) were classified to have high compliance. The mean composite score for compliance among respondents was 5.41 ± 1.5. The minimum and maximum composite scores were 2 and 8 respectively.

## Determinants of IPC compliance

Table 5 shows the results of the binary logistic regression analysis for overall IPC compliance among respondents. Females (aOR: 0.33; 95%CI: 0.17–0.64; p = 0.001) were statistically significantly less likely to exhibit high compliance. Also, nurses working in the Maternity/Labour wards (aOR: 8.31; 95%CI: 2.46–28.15; p = 0.001) and Outpatient Department (OPD)/Psychiatry (aOR: 5.00; 95%CI: 1.42–17.62; p = 0.012) were associated with higher odds of complying to IPC guidelines compared to nurses working in the Theatre/A&E/ICU. Nurses with diploma qualification (aOR: 0.48 95%CI: 0.25–0.95; p = 0.035) were less likely to have high compliance compared to those with certificate. Availability of IPC guide (aOR: 3.48; 95%CI: 1.13–10.72; p = 0.030) in a working department was associated with 3.4 the odds of exhibiting high compliance to IPC protocols.

## Discussion

This study assessed the knowledge of nurses in Yendi Municipality on IPC as well as the determinants of compliance to IPC guidelines. Our study revealed that the nurses possessed high knowledge about IPC. This finding confirms the findings of Kim and Hwang [13] in Korea, Markos et al. [22] in Ethiopia and Ziblim et al. [14] in Ghana but refutes the finding of Nofal et al. [23] among nurses and physicians in Jordan. This implies that majority of these nurses probably received adequate insight regarding IPC, through the education and training curricular of their institution. In line with the knowledge-attitude-practice-outcome model, high knowledge obtained through requisite education is expected to empower these nurses towards exhibiting good IPC practices for both their patients and themselves [24,25]. We therefore advocate that education on IPC should be regularised and sustained in all healthcare facilities.

This study also revealed that the nurses had high compliance to IPC guidelines. This finding is consistent with the report of Russell et al. [26] in North-eastern US, Mitchell et al. [27] in Australia and Ampadu [28] in Ghana. However, Geberemariyam et al. [12] in Ethiopia and Ziblim et al. [14] in Ghana reported low compliance to IPC protocols. High compliance to IPC guidelines was probably influenced by the knowledge acquired through education on IPC. High compliance could also be plausibly due to the good attitudes of these highly knowledgeable nurses towards adherence to IPC protocols [24]. Therefore, the practice of high compliance exhibited by these nurses could be a product of an interaction between knowledge and the perceived good attitude exhibited by the nurses [24,25]. Consequently, we recommend that management of healthcare facilities should put in place structures such as handwashing stations, that promote IPC and instil discipline towards adherence to such instituted IPC measures.

In this study, males were found to be statistically significantly more likely to comply with IPC guidelines. Consistent with our finding is the finding of Balegha et al. [24] among nursing students in North-western Ghana. Generally, the nursing profession is predominated by females, a phenomenon shaped by the age-long perception that the nursing profession belongs to females [29]. As Balegha et al. [24] note, male nurses in an attempt to preserve their ego and relevance are plausibly compelled to adhere to IPC guidelines. Compliance to IPC protocols

**Table 5. Determinants of IPC compliance (n = 209).**

| Variables | Low (N = 72) | High (N = 137) | cOR (95% CI) | aOR (95% CI) | P–value |
|---|---|---|---|---|---|
| | n (%) | n (%) | | | |
| **Age (years)** | | | | | |
| 20–25 | 27 (37.5) | 41 (29.9) | 1 | | |
| 26–30 | 28 (38.9) | 69 (50.4) | 1.62 (0.84–3.12) | | |
| ≥ 31 | 17 (23.6) | 27 (19.7) | 1.05 (0.48–2.28) | | |
| **Gender** | | | | | |
| Male | 29 (40.3) | 75 (54.7) | 1 | 1 | |
| Female | 43 (59.7) | 62 (45.3) | 0.56 (0.31–1.00) | 0.33 (0.17–0.64) | **0.001** |
| **Professional Rank** | | | | | |
| ≤ PEN | 24 (33.3) | 44 (32.1) | 1 | | |
| SSNM | 42 (58.3) | 80 (58.4) | 1.04 (0.56–1.94) | | |
| NMO | 6 (08.3) | 13 (09.5) | 1.18 (0.40–3.51) | | |
| **Department** | | | | | |
| Theatre/A&E/ICU | 16 (22.2) | 17 (12.4) | 1 | 1 | |
| Paed/Med/Surgical | 43 (59.7) | 62 (45.3) | 1.36 (0.62–2.98) | 1.55 (0.65–3.69) | 0.324 |
| Maternity/Labour | 7 (09.7) | 33 (24.1) | 4.42 (1.53–12.85) | 8.31 (2.46–28.15) | **0.001** |
| OPD/Psychiatry | 6 (8.3) | 25 (18.2) | 3.92 (1.28–12.05) | 5.00 (1.42–17.62) | **0.012** |
| **Qualification** | | | | | |
| Certificate | 25 (34.7) | 62 (45.3) | 1 | 1 | |
| Diploma | 43 (59.7) | 58 (42.3) | 0.54 (0.30–1.00) | 0.48 (0.25–0.95) | **0.035** |
| Degree | 4 (05.6) | 17 (12.4) | 1.71 (0.52–5.60) | 1.48 (0.35–6.21) | 0.590 |
| **Years of practice** | | | | | |
| 1–5 years | 62 (86.1) | 104 (75.9) | 1 | 1 | |
| 6–10 years | 10 (13.9) | 33 (24.1) | 1.97 (0.91–4.27) | 1.55 (0.59–4.06) | 0.371 |
| **Having prior IPC training** | | | | | |
| No | 9 (12.5) | 19 (13.9) | 1 | | |
| Yes | 63 (87.5) | 118 (86.1) | 0.89 (0.38–2.08) | | |
| **IPC committee availability** | | | | | |
| No | 13 (18.1) | 25 (18.2) | 1 | | |
| Yes | 59 (81.9) | 112 (81.8) | 0.99 (0.47–2.07) | | |
| **Availability of PPEs** | | | | | |
| No | 14 (19.4) | 28 (20.4) | 1 | | |
| Yes | 58 (80.6) | 109 (79.6) | 0.94 (0.46–1.92) | | |
| **Availability of IPC guide** | | | | | |
| No | 8 (11.1) | 8 (05.8) | 1 | 1 | |
| Yes | 64 (88.9) | 129 (94.2) | 2.02 (0.72–5.62) | 3.48 (1.13–10.72) | **0.030** |
| **Overall knowledge level** | | | | | |
| Low | 9 (12.5) | 10 (07.3) | 1 | | |
| High | 63 (87.5) | 127 (92.7) | 1.81 (0.70–4.69) | | |

[1]- Reference category, cOR; Crude Odds ratio, aOR; Adjusted Odds Ratio.

reduces the transmission of HAIs which reduces morbidity, disability and mortalities, associated with these infections among healthcare workers as well as the patients they nurse. However, a study conducted by Mitchell et al. [27] among Australian nursing students revealed no association between gender and compliance to IPC. This finding can be explained that more

males (54.7%) than females (45.3%) demonstrated high compliance to IPC protocols in this study. This probably contributed to the statistical difference observed between gender and compliance to IPC protocols in this study. Therefore, the findings of this study may not be generalisable to other populations.

Our study also revealed that nurses who worked at the maternity/labour ward and outpatient department (OPD) were more likely to exhibit high compliance to IPC guidelines. This means that department of work probably shapes the consciousness and attitude of these nurses towards complying with IPC guidelines. Nurses who work at the maternity/labour ward are required to observe high standards of asepsis when examining pregnant women. Also, nurses who work at the labour ward usually come into contact with blood and bodily fluids, which are potential sources for the transmission of infections. Therefore, it becomes only imperative that these nurses adhere to strict IPC protocols. The OPD serves as the entry point for healthcare delivery centres. Therefore, management of healthcare facilities are probably more stringent on the observance of IPC guides as it prevents the introduction and transmission of infections. Additionally, the OPD normally includes the public health department of the hospital. Therefore, public health education through posters and talks on IPC are readily accessible to these nurses. Also, an inductive analysis of our study, revealed that nurses who worked at the maternity/labour ward and OPD possessed higher knowledge on IPC compared to those who worked in other departments. Therefore, these highly knowledgeable nurses, in line with the proposition of Balegha et al. [24] and Rav-Marathe et al. [25] would be expected to show high compliance to IPC protocols. We therefore propose that, coupled with intensified health education, institutionalised IPC protocols should be enforced.

Our study also revealed that diploma nurses were statistically significantly more likely to show high compliance to IPC guidelines. However, degree nurses had low compliance to IPC protocols. This finding is in contrast with the findings of Ziblim et al. [14] in Ghana, Desta et al. [17] and Alhumaid et al. [30] in Ethiopia. This difference in compliance between the diploma and degree nurses can be explained in the context of the disproportionate sample size of the study. In our study, nurses who had diploma were in the majority (48.3%) while degree nurses constituted only 10% of the sample. Diploma holding nurses greatly outnumber the degree holders in Ghana. Therefore, this disproportionate representation of the different cadre of nurses could have influenced the association between cadre and IPC compliance. Given the nationalistic picture of the proportions of diploma nurses relative to degree nurses, the findings of this study could be generalised to other nursing populations in Ghana.

Finally, the presence of IPC guides in department where the participant/nurses worked was associated with higher likelihood of complying with the provisions of the guidelines. This finding has been reported by Geberemariyam et al. [12] and Desta et al. [17] in Ethiopia. Availability of IPC guidelines serve as the working practical framework that guides these nursing professionals towards adhering to universally established standards of IPC, to ensure that infections are prevented and controlled among healthcare workers and patients. Readily available and accessible IPC guidelines potentially increases the probability of these nurses adapting the tenets of the protocols as they work. Therefore, IPC protocols should be provided at all departments and posted at vantage points to make them readily accessible.

In this study, age, professional rank, duration of practice, having prior IPC training, availability of IPC committee, availability of PPEs and knowledge on IPC were not statistically significantly associated with compliance to IPC protocols. However, previous studies have established the association between compliance to IPC protocols and age [17,26,27], professional rank [14], duration of practice [12,13,17], prior IPC training [12,17], availability of IPC committee [12], availability of PPEs [28] and knowledge of IPC [30].

### Strengths and limitations of the study

Our study used multistage stratified random sampling technique to sample the study participants. This therefore, improves the external validity and hence representativeness of the study [24]. However, although our study administered a representative sampling technique, we cannot completely vouch for its representativeness [24]. Also, since the study was self-reported, the accuracy of the responses cannot be completely vouched for due to social desirability bias [24]. Therefore, future research should focus on more objective ways (such as a longitudinal design) of gathering information on compliance to IPC protocols.

### Conclusion

The study revealed high knowledge and compliance regarding IPC among nurses in the Yendi Municipal Hospital. A person's working department and the availability of IPC guidelines are key determinants for high compliance with IPC measures. However, knowledge on IPC was not statistically linked to its compliance. The Municipal and Regional Health directorates should therefore ensure adequate logistic flow, provision of IPC guidelines and proper supervision to ensure maximum compliance with IPC guidelines, particularly among paediatric, medical and surgical wards nurses as well as degree holding nurses

### Supporting information

**S1 Questionnaire. Knowledge and determinants of IPC compliance.**
(PDF)

**S1 Data. Data from pre-test.**
(SAV)

**S2 Data. Data from survey.**
(SAV)

### Acknowledgments

We acknowledge the expert opinion of Mr. Zakaria Abukari and Mr. Mohammed Abubakari in face and content validation of the questionnaire and Mr. Desmond Nkansah for his assistance during data collection process. The cooperation of the Yendi hospital management and respondents are duly recognized.

### Author Contributions

**Conceptualization:** Abdul-Manaf Mutaru, Raymond Kunsu.

**Data curation:** Abdul-Manaf Mutaru, Raymond Kunsu.

**Formal analysis:** Abdul-Manaf Mutaru.

**Funding acquisition:** Abdul-Manaf Mutaru.

**Investigation:** Abdul-Manaf Mutaru, Augustine Ngmenemandel Balegha.

**Methodology:** Abdul-Manaf Mutaru, Augustine Ngmenemandel Balegha.

**Project administration:** Abdul-Manaf Mutaru.

**Resources:** Abdul-Manaf Mutaru.

**Software:** Abdul-Manaf Mutaru, Collins Gbeti.

**Supervision:** Abdul-Manaf Mutaru, Raymond Kunsu, Collins Gbeti.

**Validation:** Abdul-Manaf Mutaru, Augustine Ngmenemandel Balegha, Raymond Kunsu, Collins Gbeti.

**Visualization:** Abdul-Manaf Mutaru, Augustine Ngmenemandel Balegha, Raymond Kunsu, Collins Gbeti.

**Writing – original draft:** Abdul-Manaf Mutaru, Augustine Ngmenemandel Balegha.

**Writing – review & editing:** Abdul-Manaf Mutaru, Augustine Ngmenemandel Balegha, Raymond Kunsu, Collins Gbeti.

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
