## [Decision Letter · Decision Letter 0]

10 May 2022

PONE-D-22-01831Knowledge and determinants of infection prevention and control compliance among nurses in Yendi Municipality, GhanaPLOS ONE

Dear Dr. Mutaru,

Thank you for submitting your manuscript to PLOS ONE. After careful consideration, we feel that it has merit but does not fully meet PLOS ONE’s publication criteria as it currently stands. Therefore, we invite you to submit a revised version of the manuscript that addresses the points raised during the review process.

The comments from editors and reviewers are included below, for your convenience.

We look forward to receiving your revised manuscript.

Kind regards,

Oana Săndulescu

Academic Editor

PLOS ONE

Journal Requirements:

Additional Editor Comments:

Thank you for submitting your work to PLoS One. Please find below a set of comments and queries that we feel can improve the overall presentation of the manuscript:

- The Abstracts’ methods section should specify the maximum scores calculated for knowledge, compliance, etc. This would help the readers in interpreting for example whether the 7.26 score mentioned in the Results is high or low.

- Line 32: Please mention in the text what the number after ± represents (i.e., SD)

- Line 35, 258 and table legends: Please define the OPD abbreviation.

- Lines 36, 40, and 350: Phrasing should be refined, since it currently sounds a bit odd. Of course a guideline should exist in order for it to be followed. The full text manuscript clarifies on line 119 that this refers to whether or not specific IPC guidelines existed in the department. This clarification should also be added to the Abstract.

- Line 66: Please revise the term “modest”.

- Overall, the manuscript’s introduction would benefit from more structuring, to focus only on HAIs. Right now, there is a mix, for example, line 75 mentions communicable diseases in general, and lists COVID-19 as example, without mentioning whether there is any connection between this phrase and the rate of COVID HAIs.

- Line 86: Please revise this statement: “This study employed a descriptive quantitative cross-sectional study” – this appears to have been a cross-sectional survey. Please avoid repeating the term “study” and please revise the use of the term “quantitative”.

- What is the relevance of including lines 94-95 in the Methods section? They could at best be used to bring an argument as to why this study was needed in the Introduction section, or in the Discussion section, but they are not a good fit for the Methods section.

- Line 107: Please define: Theater.

- Line 174: Please correct: “date” to “data”.

- Please start the Results section by specifying the rate of non-response in total and by department/level of qualification.

- Line 207: Please correct “a thirds” to “one third”.

- Line 207-208: There probably is an error in phrasing here, and you were referring to the percentage of respondents who correctly reported that recapping should not be performed, as can be deduced from Table 2.

- Line 233: Please correct “rob” to “rub”.

- Line 281: Phrasing is not very clear, please elaborate: “right attitudinal posture”.

- Line 284: What is the connection with students?

- Line 308: Please revise the term “harbors”.

- Line 330: Please rephrase “pasted”.

- Throughout the discussion, there is a repetition that further studies should be conducted, i.e., lines 296-297 and 321-322, and line 347. I would suggest that instead of comparing the results so much with field literature, you focus on highlighting on what your findings mean for your specific setting and how these could be generalized, or not, to a wider setting in your country.

- Lines 339-341 would be a better fit in the Methods section.

- Line 353: Please check spelling for guidelines.

- Lines 361-364: Please confirm that all persons mentioned here have consented to have their name published.

- Please check that all the information in the supplementary files provided is anonymized and cannot be considered as identifiable data.

Reviewers' comments:

Reviewer's Responses to Questions

**Comments to the Author**

1. Is the manuscript technically sound, and do the data support the conclusions?

Reviewer #1: Yes

Reviewer #2: Yes

2. Has the statistical analysis been performed appropriately and rigorously? 

Reviewer #1: Yes

Reviewer #2: Yes

3. Have the authors made all data underlying the findings in their manuscript fully available?

Reviewer #1: Yes

Reviewer #2: Yes

4. Is the manuscript presented in an intelligible fashion and written in standard English?

Reviewer #1: Yes

Reviewer #2: Yes

5. Review Comments to the Author

Reviewer #1: Knowledge and determinants of infection prevention and control compliance among nurses in Yendi Municipality, Ghana

Reviewer’s comments

This study assessed the knowledge and determinants of infection prevention and control (IPC) compliance among nurses in northern zone of Ghana. This is important especially in the wake of the COVID-19 pandemic and recent outbreaks of diseases such as Ebola virus disease, Lassa Fever in the countries in the West African sub region since IPC practices are crucial for controlling the transmission of these diseases.

The manuscript is fairly well-written with a clear focus on the objective of the study. The authors may consider addressing the comments below to improve on the manuscript.

1. Abstract – In the methods section (line 29, 30), the authors should consider adding that the multivariate logistic regression was used.

2. Introduction- line 49- consider inserting ‘associated with high’ before the expenditure if it explains better what the authors intend to communicate.

3. Materials and methods-

• It is not clear what the inclusion and exclusion criteria are.

• It is unclear whether Line 95 is referring to the hospital or the entire municipality. The authors should consider specifying where exactly the inadequacy/lack of resources is referring to.

• A summary of the numbers obtained for the different cadre of professionals (from the various units) after the selection based on the probability proportional to strata approach for selection would be helpful.

4. Data analysis-

• Consider rephrasing line 133-134 – to ‘Frequencies and percentages were used to summarize data on the socio-demographic characteristics, knowledge and compliance to IPC guidelines.

• Line 140 – There is no need to include median (with IQR) since the authors ascertained that the data was normally distributed.

• The authors would need to explain why statistical significance was set at p<0.20 at the bivariate analysis stage.

• Consider rephrasing line 150 as it is currently confusing. The previous comment above may provide a guide.

• Line 163 – Cronbach is a name so should start with upper case.

5. Results

• Table 1

o The unit of age, i.e. years, should be included in the table

o The NMO should also be explained in the footnote

• Line 207 – Consider starting the sentence with ‘Less’.

• Line 207- it is unclear the type of recapping the authors are describing as ‘aptly reported’. The authors should bear in mind that if the recapping is done holding the used needle and the cap in one’s hands, it would be very dangerous and an inappropriate IPC practice. This needs further clarification.

• Table 2 – The ‘No’ column is implied since those are the only two options. The authors should consider deleting the column and presenting the % separately as was done with Table 1.

• Line 231 – According to the WHO, the recommended duration of the entire handwashing procedure should be 40-60 seconds. In this study, the authors seem to have considered 1 minute (60 seconds) as the ideal. The researchers would need to provide justification (and reference) for this choice.

• Table 3 – Similar comment as for Table 2.

• Consider describing the content of Table 4 in prose only and deleting Table 4.

• The authors should consider reconciling the terminologies used in the write up on the data analysis with that used in the write up on the overall knowledge level and compliance (lines 241-246) - good versus High level; poor versus low

• Line 260-261 – The authors should consider explaining the different nursing training and qualifications so that the global readers can understand when ‘diploma or certificate’ is used. This can be explained in the study population section.

6. Discussion

• Line 270-271. The reasons given may or may not be the case and the authors should consider rephrasing to reflect this.

• Line 276-279. The authors should consider rewording the statement so that the current study findings are the focus before any comparisons are carried out, as was done in lines 267-270.

• Line 284- 285. Consider rephrasing to reflect the need for structures to promote the adherence to IPC protocols.

• Line 293 – Consider inserting ‘of HAIs’ after transmission and ‘these’ before infections.

• Lines 323. Consider using ‘department where the participant/nurses worked’ instead of ‘working department’.

• Strengths and limitations

o Lines 341 – 343. Consider rephrasing the sentence and deleting the phrase on what multistage sampling is.

o Line 344 – the authors should provide information on what the power of this study is.

o Line 347- consider ending the sentence with ‘as well as more objective ways of gathering information on compliance to IPC protocols.’

7. References

• Check and update references according to the journal’s guidelines. Some of the references are incomplete.

• Also check reference 10 and 16

Reviewer #2: The authors have adhered to all principles of publication by going through the headings: Abstract, Introduction, Methods Sampling, Data collection, Data analysis, Validity and Reliability Ethics and results.

In terms of sample size calculation the authors used Yamane's formula, which assumes a prevalence of 50% as that of the main outcome. Authors should provide the unit of AGE (years) in the tables (line 199 and 264)

6. PLOS authors have the option to publish the peer review history of their article (what does this mean?). If published, this will include your full peer review and any attached files.

Reviewer #1: No

Reviewer #2: **Yes: **Martin Adjuik

---

## [Author Response · Author response to Decision Letter 0]

21 May 2022

Academic Editor

we are have implemented all your directions in to the manuscript. It has been very helpful. thank you for the comments.

Reviewer 1:

we have incorporated all your comments which has been very helpful in rebuilding the manuscript. we are very grateful for your review.

Reviewer 2:

we have considered all your inputs and implemented them. we thank you so much for the comments.

---

## [Decision Letter · Decision Letter 1]

12 Jun 2022

Knowledge and determinants of infection prevention and control compliance among nurses in Yendi Municipality, Ghana

PONE-D-22-01831R1

Dear Dr. Mutaru,

We’re pleased to inform you that your manuscript has been judged scientifically suitable for publication and will be formally accepted for publication once it meets all outstanding technical requirements.

Kind regards,

Oana Săndulescu

Academic Editor

PLOS ONE

Additional Editor Comments (optional):

I thank the authors for addressing all previous review comments.

Reviewers' comments:

Reviewer's Responses to Questions

**Comments to the Author**

1. If the authors have adequately addressed your comments raised in a previous round of review and you feel that this manuscript is now acceptable for publication, you may indicate that here to bypass the “Comments to the Author” section, enter your conflict of interest statement in the “Confidential to Editor” section, and submit your "Accept" recommendation.

Reviewer #1: All comments have been addressed

2. Is the manuscript technically sound, and do the data support the conclusions?

Reviewer #1: Yes

3. Has the statistical analysis been performed appropriately and rigorously? 

Reviewer #1: Yes

4. Have the authors made all data underlying the findings in their manuscript fully available?

Reviewer #1: Yes

5. Is the manuscript presented in an intelligible fashion and written in standard English?

Reviewer #1: Yes

6. Review Comments to the Author

Reviewer #1: (No Response)

7. PLOS authors have the option to publish the peer review history of their article (what does this mean?). If published, this will include your full peer review and any attached files.

Reviewer #1: No

---

## [Editor Report · Acceptance letter]

27 Jun 2022

PONE-D-22-01831R1 

Knowledge and determinants of infection prevention and control compliance among nurses in Yendi municipality, Ghana. 

Dear Dr. Mutaru:

I'm pleased to inform you that your manuscript has been deemed suitable for publication in PLOS ONE. Congratulations! Your manuscript is now with our production department. 

Kind regards, 

on behalf of

Dr. Oana Săndulescu 

Academic Editor

PLOS ONE